# Do Current Smokers and Ex-Smokers Who Use Nicotine Vaping Products Daily Versus Weekly Differ on Their Reasons for Vaping? Findings from the 2020 ITC Four Country Smoking and Vaping Survey

**DOI:** 10.3390/ijerph192114130

**Published:** 2022-10-29

**Authors:** Shannon Gravely, Hua-Hie Yong, Jessica L. Reid, Katherine A. East, Coral E. Gartner, David T. Levy, K. Michael Cummings, Ron Borland, Anne C. K. Quah, Maansi Bansal-Travers, Janine Ouimet, Geoffrey T. Fong

**Affiliations:** 1Department of Psychology, University of Waterloo, 200 University Avenue West, Waterloo, ON N2L 3G1, Canada; 2School of Psychology, Deakin University, 1 Gheringhap Street, Geelong, VIC 3220, Australia; 3School of Public Health Sciences, University of Waterloo, 200 University Avenue West, Waterloo, ON N2L 3G1, Canada; 4Department of Addictions, Institute of Psychiatry, Psychology & Neuroscience, King’s College London, Addictions Sciences Building, 4 Windsor Walk, Denmark Hill, London SE5 8BB, UK; 5School of Public Health, Faculty of Medicine, The University of Queensland, The Public Health Building, Corner of Wyndham St and Herston Road, Herston, QLD 4006, Australia; 6Lombardi Comprehensive Cancer Center, Georgetown University, 3800 Reservoir Rd NW, Washington, DC 20007, USA; 7Department of Psychiatry & Behavioral Sciences, Medical University of South Carolina, 67 President St., Charleston, SC 29425, USA; 8School of Psychological Sciences, University of Melbourne, Grattan Street, Parkville, VIC 3010, Australia; 9Department of Health Behavior, Roswell Park Comprehensive Cancer Center, Elm and Carlton Streets, Buffalo, NY 14263, USA; 10Ontario Institute for Cancer Research, 661 University Ave., Toronto, ON M5G 0A3, Canada

**Keywords:** nicotine vaping products, cigarettes, vaping, smoking, use reasons, adults

## Abstract

This study examined reasons why adults who currently smoke or formerly smoked cigarettes use nicotine vaping products (NVPs) by vaping frequency (daily vs. weekly) stratified by smoking status. This cross-sectional study included 3070 adults from the 2020 ITC Four Country Smoking and Vaping Survey (Australia, Canada, England, United States) who reported using a NVP (vaping) at least weekly and who either currently smoke (*n* = 2467) or formerly smoked (*n* = 603). Respondents were asked to select the reason(s) they use NVPs, including to manage their smoking (reduce/quit or remain quit) and/or for reasons unrelated to managing smoking (e.g., to save money, enjoyment, flavours). We found that both current and former smokers endorsed an average of six reasons for vaping, with those vaping daily reporting significantly more reasons than those vaping weekly. Among current smokers, 72.8% reported vaping may help them quit smoking, 13.0% reported vaping to reduce smoking but not to quit, and 14.2% reported vaping only for reasons other than to reduce or quit smoking. The most common reason for vaping among current smokers was to reduce smoking (81.3%). Current smokers vaping daily were significantly more likely than those vaping weekly to report using a NVP to reduce smoking, for enjoyment, to reduce harm to themselves and others, to quit smoking, likeable flavours, and to save money. The most common reason cited for vaping by respondents who formerly smoked was enjoyment, with those who vaped daily more likely than those who vaped weekly to report vaping for enjoyment and to reduce harm to themselves. Nearly all reported vaping to help stay abstinent from smoking (92.3%), with no significant difference by vaping frequency. In conclusion, a majority of respondents reported using NVPs to manage their smoking (reduce/quit smoking or remain quit), particularly those vaping daily. Those who were vaping daily also endorsed a greater number of reasons other than managing smoking relative to those who were vaping weekly.

## 1. Introduction

Combustible tobacco products, predominantly cigarettes, cause the majority of tobacco-related illnesses and deaths [1,2]. Although cigarette smoking prevalence has decreased significantly over the past several decades, globally, an estimated 1 billion people still smoke cigarettes [3]. While the benefits of quitting smoking are well-documented [4,5,6], and many nicotine-dependent smokers have tried to quit [6,7], few manage to quit successfully [4,7,8,9,10]. Moreover, around one-third who try to quit smoking do so unassisted, rather than utilizing government-approved cessation aids or other forms of assistance [11].

Over the last decade, non-combustible alternative nicotine delivery systems, including nicotine vaping products (NVPs, also known as e-cigarettes), have entered the global market and have been noted for their potential as a possible substitute for cigarettes, delivering nicotine at lower harm [12]. Scientific reviews have concluded that although NVPs are not harmless, completely switching from cigarettes to NVPs greatly reduces exposure to several toxicants, including carcinogens [13,14,15,16]. Biomarker studies also indicate that NVPs expose people to fewer harmful substances than cigarettes, which are associated with cancer, respiratory conditions, and cardiovascular diseases [15]. Additionally, there is ‘moderate-certainty’ evidence that NVPs help some people to quit smoking cigarettes [16], particularly if NVPs are used more frequently (e.g., daily) [17,18,19]. Notably, the self-reported use of NVPs for smoking cessation has surpassed the use of other government-approved medical therapies in some countries [11,20,21].

Reasons for using NVPs vary among adults who smoke cigarettes or who have quit smoking [22]. While many adults report using NVPs to quit smoking, other reported motivations include saving money, reducing (but not stopping) smoking, perceived lower risk than cigarettes, greater social acceptability, and the ability to vape where smoking is not allowed [22,23,24,25]. Thus, as the potential harm reduction benefit of NVPs may depend on whether they are used to stop smoking cigarettes or to remain abstinent from smoking, there is a need to understand the reasons for using NVPs among those who currently smoke or formerly smoked.

Yong et al. [22], analyzed data from the 2016 (Wave 1) International Tobacco Control Policy Evaluation Project Smoking and Vaping Survey (ITC 4CV) in four countries (United States (US), Canada, England, and Australia) and found that among adults who currently smoke, the most commonly reported reasons for regular vaping were to reduce smoking, to quit smoking, and because vaping is less harmful to others than smoking. Among those who formerly smoked, the most common reasons for vaping were for enjoyment, because vaping is less harmful to others, greater affordability, and to help them stay abstinent from smoking. Many reasons for vaping differed by smoking and vaping frequency (daily vs. weekly), by country, and age.

Several market and policy changes have occurred in the four studied countries since the 2016 data were reported by Yong et al. [22]. For example, Canada regulated NVPs in 2018 [26], thus expanding product availability and retail access. Additionally, Health Canada’s Tobacco Strategy stated that NVPs could be helpful for smokers attempting to quit, particularly if they were unsuccessful with other medically approved cessation aids [27]. In February 2020, the US FDA prioritized enforcement efforts against flavoured pre-filled cartridge/pod NVP devices, with the exception of tobacco and menthol flavours [28]. In contrast, few changes occurred in Australia, which continued to prohibit the sale of NVPs, unless on prescription from a doctor for an unapproved medicine [29], or in England, where public health organizations (e.g., Public Health England) maintained their position that NVPs can assist smoking cessation [30], and where the National Institute for Health and Care Excellence (NICE) clinical practice guidelines included NVPs as an option for assisting with smoking cessation [31].

The current study, reporting on ITC 4CV data from 2020, provides an update from Yong et al. [22] on reasons for using NVPs among adults who currently and previously smoked by vaping frequency (daily vs. weekly) across Australia, Canada, England, and the US. Given that NVPs regulations and support for their use for quitting smoking differs across countries, we also assessed whether for people currently smoking, there were country differences in NVP use for smoking cessation purposes as opposed to for smoking reduction only (without cessation), or only reasons not related to reducing smoking or for smoking cessation purposes.

## 2. Materials and Methods

### 2.1. Study Design, Procedure, and Population

The ITC 4CV Survey is a cohort study of parallel online surveys conducted in Canada, the US, England, and Australia. Respondents (adults ≥ 18 years) were recruited by commercial panel firms in each country as established cigarette smokers (smoke ≥ monthly, and smoked at least 100 cigarettes in their lifetime), recent ex-smokers (quit smoking ≤ 2 years), or nicotine vapers (vape ≥ weekly). The sample in each country was designed to be as representative as possible of each of these groups (e.g., by age, gender, and region). All data were collected online, and respondents were remunerated. Further methodological information can be found in the technical reports [32,33,34] and Thomson et al. [35].

The current study used cross-sectional data from the ITC 4CV Wave 3 Survey (February-June 2020). ITC 4CV3 respondents included those retained in previous waves (4CV1 and/or 4CV2) and new respondents recruited to replace those lost to follow-up. Eligible respondents for this study included those who vaped at least weekly (those vaping less than weekly were not asked the outcome questions relevant to this study) and who also smoked cigarettes at least weekly or who had quit smoking. A total of 11,607 respondents completed the 2020 4CV3 Survey. Of those, 3810 reported vaping at least weekly (e.g., every day, some days of the week). We excluded respondents who were smoking less than weekly (*n* = 245) and had never smoked (*n* = 495), resulting in 3070 adults who either smoked cigarettes (*n* = 2476; referred to herein as ‘current smokers’) or had quit smoking (*n* = 603; quit smoking 0–12 months ago: *n* = 175; 13–18 months ago: *n* = 100; 2+ years ago: *n* = 328; referred to herein as ‘ex-smokers’). A study flow diagram is shown in Appendix A. Further details about the ITC 4CV3 study are detailed elsewhere [34].

### 2.2. Measures

The 4CV3 survey is available at: https://itcproject.org/surveys/ (accessed on 27 October 2022). The following variables were used in the current study:

#### 2.2.1. Independent Variable: Vaping Frequency

Respondents were asked: “How often, if at all, do you currently use vaping products (i.e., vape)?” Those who reported vaping daily or weekly were included in this study (respondents who vaped less than weekly did not receive the survey question about reasons for vaping). This variable was categorized as “daily vaping” or “weekly vaping”.

#### 2.2.2. Covariates

##### Smoking Status and Frequency

Respondents were asked: “How often, if at all, do you currently smoke ordinary cigarettes (either factory-made/packet or roll-your-own)?” Those who reported smoking daily or weekly and those who had quit smoking (recent quitters: ≤2 years ago vs. longer-term quitters: >2 years ago) were included.

##### Sociodemographics

Age group (18–39 vs. 40+), sex (male vs. female), income (low, moderate, high, or not reported), education (low, moderate, high, or not reported), and country of residence were collected by the online panel firms and verified by each of the respondents at the time of survey completion.

#### 2.2.3. Outcome Measures

Respondents were asked: “Select all that apply: Which of the following are reasons that you use vaping products?”, with nine possible reasons for current smokers and eight for ex-smokers. Each reason was coded as ‘yes’ (selected) vs. ‘no’ (not selected). Next, as per Yong et al. [22], for current smokers, a three-category quit–reduce composite measure was derived and coded as: (1) those who selected to quit smoking as a reason for vaping; (2) those who selected to reduce smoking but not to quit smoking; or (3) those who selected neither to reduce or quit smoking. The outcomes (reasons for vaping and the quit-reduce composite variable) are described in Box 1.

Box 1ITC 4CV3 Survey questions assessing reasons for vaping and outcome measure coding
**Reasons for Vaping: Survey Question**

**Outcome: Coding**

**Asked of Smokers/Ex-Smokers**
Vaping is less harmful to me than smoking(0) no vs. (1) yesBothVaping is less harmful than smoking to other people around meBothI enjoy vaping BothI save money by vaping instead of smokingBothI like the e-liquid flavoursBothVaping is more acceptable than smokingBothI can vape in places where I can’t smokeBothVaping helps me cut down on the number of cigarettes I smokeSmokers onlyVaping might help me stop smoking ordinary cigarettesSmokers onlyVaping might help me stay quit from smoking cigarettesEx-smokers onlyComposite quit-reduce measure(1) selected quitting smoking as a reason for vaping; (2) selected reducing number of cigarettes but did not select quitting smoking; (0) did not select reducing number of cigarettes or quitting smoking (selected other reasons only)Smokers only

### 2.3. Statistical Analysis

Sample characteristics were examined using frequencies and unweighted percentages. All other analyses were conducted on weighted data (using calibrated cross-sectional sampling weights). In brief, a raking algorithm was used to calibrate the weights on smoking and vaping status, geographic region, and demographic measures. Weighting also adjusts for oversampling of vapers and younger respondents (aged 18–24). The weight calibration used benchmarks from national surveys from each of the respective countries [34,36]. Statistical significance and confidence intervals were computed at the 95% confidence level. Analyses were conducted in SAS Version 9.4.

The first set of analyses included separate adjusted logistic regression analyses (proc surveylogistic) for each of the possible reasons given by respondents for vaping (yes vs. no; 9 models for current smokers and 8 for ex-smokers). Each model was stratified by smoking status to: (1) estimate the proportion of respondents who selected each reason for using NVPs; and (2) test differences by vaping frequency (daily vs. weekly NVP use). We also examined the mean number of reasons selected by smoking status. The primary independent variable was vaping frequency. Models that included current smokers adjusted for age group (18–39 vs. 40), sex (male vs. female), income (low, moderate/high, not reported), education (low, moderate/high, not reported), country, and smoking frequency (daily vs. weekly). For ex-smokers, models were adjusted for age group, sex, income, education, country, and quit duration (quit ≤2 years ago vs. >2 years ago).

The second set of analyses assessed the composite quit-reduce measure among current smokers using a multinomial regression analysis. Country of residence was the independent variable. The outcome variable was categorized as: (1) those who selected quitting smoking as a reason for vaping; (2) those who selected reducing number of cigarettes but did not select quitting smoking; or (3) those who selected neither cigarette reduction nor quitting smoking. Estimates were derived overall and by country, adjusting for age group, sex, income, and education. We also identified whether any of the covariates in the model were significantly related to using NVPs to quit smoking. We could not include smoking frequency as a covariate in the model due to convergence issues resulting from the small number of weekly smokers. A post hoc analysis was used to examine differences between countries.

## 3. Results

### 3.1. Sample Characteristics

Table 1 presents the unweighted descriptive statistics of the respondents in this study
sample. In brief, 56.7% of respondents were male, 61.3% were between the ages of 18 and 39, 73.4% reported having a moderate/high annual household income, 79.8% had a moderate/high level of education, and 59.5% were current daily smokers (19.7% were former smokers).

### 3.2. Study Results

Table 2 presents the reasons for vaping among current smokers and ex-smokers by vaping frequency. A qualitative comparison of the 2016 [22] and 2020 (current study) ITC data on reasons for using NVPs are presented in Appendix A. 

#### 3.2.1. Reasons for NVP Use among Current Smokers

Figure 1 present reasons for vaping among current smokers by vaping frequency. Results for the (full) adjusted models on reasons for vaping among current smokers can be found in Appendix A.The three most common reasons for vaping reported by current smokers were: to reduce smoking (81.3%), for enjoyment (78.2%), and because vaping is less harmful than smoking to others (74.5%). Out of the possible nine options, the mean number of reasons for vaping selected by current smokers was 6.38 (standard deviation = 2.32). Daily vapers selected a greater mean number of reasons [6.72 (SD = 2.27)] than weekly vapers [5.99 (SD = 2.31)] (*p* < 0.001).

Current smokers who vaped daily were significantly more likely than those who vaped weekly to report vaping for enjoyment, to save money, to reduce smoking, to quit smoking, likeable (e-liquid) flavours, and vaping is less harmful than smoking to themselves and to others (see Table 2).

#### 3.2.2. Vaping for Smoking Reduction/Cessation versus Other Reasons among Current Smokers, Overall and by Country

Figure 2 presents results from the quit-reduce composite analysis. Overall, 72.8% of current smokers reported using NVPs to quit smoking, 13.0% reported vaping to reduce smoking but not to quit, and 14.2% reported vaping for other reasons only (not to reduce or quit smoking). There were some differences between countries: US respondents were significantly less likely to select quitting smoking as a reason for vaping (68.0%) compared to Canada (75.2%) and Australia (79.2%). Respondents in Australia were less likely to report vaping for other reasons only (8.4%) compared to those in Canada (11.7%) and the US (14.5%). Respondents from England did not differ significantly from the other three countries.

Some model covariates were associated with the outcome (quitting smoking vs. vaping for other reasons only, results are not shown in the table): those aged 18–30 were significantly less likely to report vaping to quit smoking (67.7%) compared to those aged 40+ (78.3%, *p* < 0.001); and, current smokers who vaped daily were more likely to report vaping to quit smoking (76.1%) relative to those vaping weekly (68.8%, *p* = 0.006). Sex trended towards significance for vaping to quit smoking (females: 75.8% vs. males: 70.5%, *p* = 0.051).

#### 3.2.3. Reasons for NVP Use among Ex-Smokers

Figure 3 present reasons for vaping among ex-smokers by vaping frequency. Results for the (full) adjusted models on reasons for vaping among ex-smokers can be found in Appendix A.

The three most common reasons for vaping reported by ex-smokers were: for enjoyment (93.7%), vaping is less harmful to themselves than smoking (92.7%), and vaping may help them stay abstinent from smoking (92.3%). Out of the eight possible options, the mean number of reasons was 6.38 (1.54). Daily vapers selected a greater mean number of reasons 6.48 (1.43) than weekly vapers [5.79 (1.98)] (*p* < 0.001).

Ex-smokers who vaped daily were significantly more likely than those who vaped weekly to report vaping because it is less harmful to themselves than smoking, and for enjoyment. There were no differences between those who vaped daily vs. weekly for the other six reasons (see Table 2). 

## 4. Discussion

This study examined reasons for using NVPs among adults who currently or formerly smoked, by vaping frequency. We found that the most common reason for vaping among current smokers was to reduce smoking. A majority of current smokers reported vaping might help them quit smoking (about 7 in 10), with only a small proportion (about 1 in 10) reporting vaping to merely reduce smoking but not to quit. Nearly all ex-smokers reported vaping because it may help them remain abstinent from smoking, regardless of their quit duration and vaping frequency. There were several differences in reasons for vaping among current smokers who vaped more frequently (daily) than those vaping less frequently (weekly); however, there were fewer differences among ex-smokers by vaping frequency.

The results were generally similar to the 2016 findings reported by Yong et al. [22]. For example, among current smokers, similar proportions of respondents reported vaping to reduce smoking (2016: 86% vs. 2020: 81%), to quit smoking (2016: 77% vs. 2020: 73%), because they perceive vaping as less harmful to others (2016: 78% vs. 2020: 75%), and because they perceive that vaping is more acceptable than smoking (2016: 74% vs. 2020: 70%). In 2020, slightly more smokers reported vaping for enjoyment than in 2016 (70% vs. 78%), suggesting that the quality of NVPs are improving (e.g., more brand and/or flavour options, more effective nicotine delivery systems). Additionally, compared to 2016, we found the proportion of current smokers who reported using NVPs for reasons other than quitting was similar (2016: 23% vs. 2020: 27%); however, a larger proportion of smokers in 2020 were vaping to cut down, rather than to quit than in 2016 (2016: 6% vs. 2020: 13%); however, this was still somewhat negligible in 2020. Among ex-smokers, a slightly higher proportion reported vaping to stay abstinent from smoking in 2020 than in 2016 (2016: 88% vs. 2020: 92%), but fewer ex-smokers reported vaping because it is more acceptable than smoking (2016: 79% vs. 2020: 66%), suggesting that norms towards vaping are becoming more negative over time. This may be in part due to an increasing proportion of the general adult population who believe that NVPs are as harmful or more harmful than cigarettes, particularly in the US [37,38,39]. All other reasons were similar between survey years.

Both current and ex-smokers endorsed multiple reasons for vaping; however, those who vaped daily reported significantly more reasons than those who vaped weekly. This is consistent with the notion that non-daily use is more experimental and/or situationally cued rather than for reasons related to managing their smoking and/or for product likeability and satisfaction relative to smoking [40]. The high number of reasons for vaping among daily vapers is likely because of their positive experiences with NVPs, and satisfaction with vaping compared to smoking. Smokers may initially try a vaping product for smoking cessation purposes, and may continue to vape more regularly for additional reasons if they like the product and find multiple benefits from using them.

There were some differences between countries among current smokers using NVPs. Respondents in the US were less likely to report vaping to quit smoking than those in Canada and Australia. Additionally, fewer Australian respondents reported vaping for reasons other than to reduce/quit smoking, which may reflect the policy environment and limited access to NVPs. England did not show a higher proportion of NVP use for cessation purposes relative to the other three countries, which is surprising considering that NVPs are supported as a cessation aid in NICE clinical practice guidelines [31], but absent from guidelines in the other countries. However, interest in use NVPs to quit smoking was high across all four countries.

While this study has many strengths, including the large sample size spanning four countries, there are limitations to consider. First, this study is cross-sectional; thus, we do not attempt to imply causality (e.g., whether reasons for using NVPs preceded the decision to use them). Second, respondents could choose multiple reasons for vaping; thus, the primary reason(s) are unknown. Third, this was not an exhaustive list of possible reasons for vaping; thus, there could be other reasons that also may also be important. Fourth, the generalizability of findings is limited to only those who vape frequently (at least weekly) and who were currently smoking or had quit smoking. External validity beyond the four counties studied may also be limited, especially for reasons that are more socially influenced.

## 5. Conclusions

In all four countries, most respondents reported that NVPs may help them quit smoking or remain abstinent as a reason for vaping, particularly among those vaping daily. Those who vape daily also appear to endorse a greater variety of other reasons for using NVPs.

## Figures and Tables

**Figure 1 ijerph-19-14130-f001:**
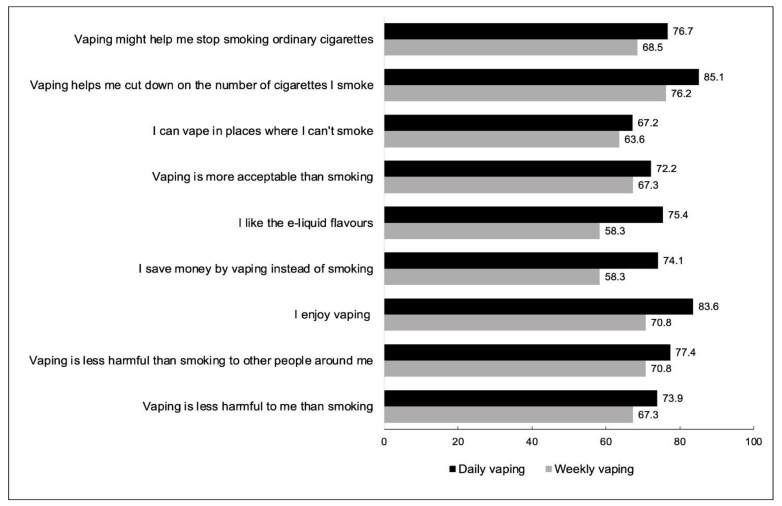
Reasons for vaping among current smokers by vaping frequency, (% yes, weighted).

**Figure 2 ijerph-19-14130-f002:**
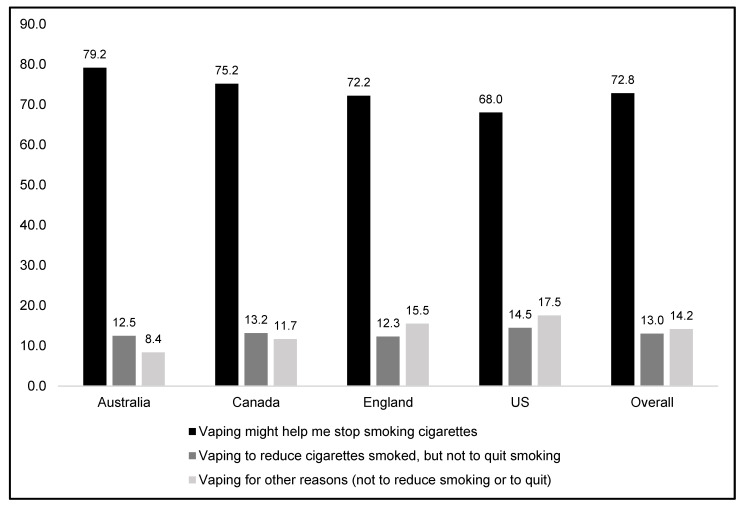
Reasons for vaping among current smokers: quit-reduce smoking versus other reasons only, (% yes, weighted)**.** N = 2467. Data are weighted and adjusted for sex, age, income, education, and vaping frequency.

**Figure 3 ijerph-19-14130-f003:**
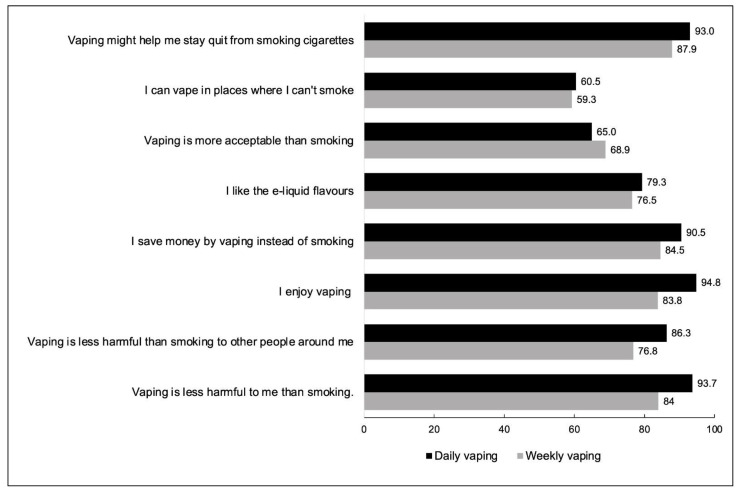
Reasons for vaping among ex-smokers by vaping frequency (% yes, weighted).

**Table 1 ijerph-19-14130-t001:** Respondent characteristics (*n*, unweighted %) of the study sample, Wave 3 (2020).

	All Respondents N = 3070		Daily NVP Use *n* = 1824 (59.4%)	Weekly NVP Use *n* = 1246 (40.6%)
	*n*	%	%	%
**Country of residence**				
Australia	202	6.6	7.0	5.9
Canada	927	30.2	25.8	36.6
England	1300	42.4	44.4	39.4
United States	641	20.9	22.8	18.1
**Sex**				
Male	1741	56.7	55.8	58.0
Female	1329	43.3	44.2	42.0
**Age**				
18–39	1882	61.3	57.7	66.5
40+	1188	38.7	42.3	33.5
**Income**				
Low	723	23.6	24.6	22.0
Moderate	861	28.1	27.8	28.4
High	1390	45.3	44.4	46.6
Not reported	96	3.1	3.2	3.1
**Education**				
Low	593	19.3	21.5	16.1
Moderate	1350	44.0	43.5	44.7
High	1100	35.8	33.9	38.6
Not reported	27	0.9	1.1	0.6
**Smoking status**				
Daily smoking	1828	59.5	57.3	62.8
Weekly smoking	639	20.8	14.3	30.3
Former smoking (quit < 2 years)	275	9.0	11.7	5.0
Former smoking (quit ≥ 2 years)	328	10.7	16.7	1.9

Data are unweighted and unadjusted. Income is defined as: ‘low’ (CA: <CAD $30,000; US: <USD $30,000; AU: <AUD $30,000; EN: <£15,000), ‘moderate’ (CA: CAD $30,000–59,000; US: USD$30,000–59,000; AU: AUD $30,000–59,000; EN: £15,000–30,000), ‘high’ (CA: ≥CAD $60,000; US: ≥USD $60,000; AU: ≥AUD $60,000; EN: >£30,000), and ‘not reported’; Education is defined as: ‘low’ (all countries: ≤ high school), ‘moderate’ (CA: trade school, community college, some university (no degree); US: trade school, community college, associate degree, or some university (no degree); AU: technical education or some university (no degree); EN: further education/training college below degree level or some university (no degree), ‘high’ (all countries: university degree or post-graduate degree), and ‘not reported’.

**Table 2 ijerph-19-14130-t002:** Reasons for vaping (% yes, weighted) by vaping frequency and smoking status.

	Current Smokers			Ex-Smokers		
Reasons for Vaping, % Yes Weighted (95%CI)	Daily Vaping (*n* = 1306)	Weekly Vaping (*n* = 1161)	Total (*n* = 2467)	Daily Vaping (*n* = 518)	Weekly Vaping (*n* = 85)	Total (*n* = 603)
Vaping is less harmful to me than smoking	73.9 (70.2–77.3)	67.3 (63.5–70.9)	70.9 (68.2–73.4)	93.7 (90.1–96.1)	84.0 (71.2–91.7)	92.7 (89.1–95.1)
aOR (95% CI)	1.37 (1.07–1.77)	Reference		2.85 (1.20–6.76)	Reference	
Vaping is less harmful than smoking to other people around me	77.4 (74.0–80.6)	70.8 (67.0–74.4)	74.5 (71.9–76.9)	86.3 (80.3–90.6)	76.8 (62.6–86.7)	85.0 (79.8–89.1)
aOR (95% CI)	1.41 (1.09–1.83)	Reference		1.90 (0.81–4.47)	Reference	
I enjoy vaping	83.6 (80.5–86.4)	70.8 (67.0–74.4)	78.2 (75.8–80.5)	94.8 (92.0–96.6)	83.8 (69.4–92.2)	93.7 (90.9–95.7)
aOR (95% CI)	2.10 (1.58–2.80)	Reference		3.50 (1.34–9.10)	Reference	
I save money by vaping instead of smoking	74.1 (70.6–77.2)	58.3 (54.2–62.3)	67.0 (64.3–69.6)	90.5 (86.3–93.6)	84.5 (72.5–91.9)	89.7 (85.7–92.7)
aOR (95% CI)	2.04 (1.61–2.60)	Reference		1.76 (0.78–3.96)	Reference	
I like the e-liquid flavours	75.4 (71.9–78.6)	67.5 (63.6–71.3)	71.9 (69.3–74.3)	79.3 (73.8–84.0)	76.5 (60.3–87.4)	78.9 (73.8–83.3)
aOR (95% CI)	1.48 (1.14–1.90)	Reference		1.18 (0.52–2.69)	Reference	
Vaping is more acceptable than smoking	72.2 (68.6–75.6)	67.3 (63.4–71.0)	69.9 (67.3–72.5)	65.0 (58.5–70.9)	68.9 (53.3–81.1)	65.6 (59.8–70.9)
aOR (95% CI)	1.26 (0.99–1.61)	Reference		0.84 (0.40–1.75)	Reference	
I can vape in places where I can’t smoke	67.2 (63.5–70.8)	63.6 (59.6–67.4)	65.5 (62.8–68.2)	60.5 (54.1–66.5)	59.3 (43.5–73.3)	60.3 (54.4–65.9)
aOR (95% CI)	1.17 (0.93–1.48)	Reference		1.05 (0.53–2.10)	Reference	
**Smoking Management**						
Vaping helps me cut down on the number of cigarettes I smoke	85.1 (82.2–87.6)	76.2 (72.3–79.6)	81.3 (78.9–83.5)	—	—	—
aOR (95% CI)	1.79 (1.34–2.38)	Reference				
Vaping might help me stop smoking cigarettes	76.7 (73.2–79.8)	68.5 (64.5–72.2)	73.0 (70.4–75.4)			
aOR (95% CI)	1.51 (1.17–1.96)	Reference		—	—	—
Vaping might help me stay quit from smoking cigarettes	—	—	—	93.0 (89.3–95.4)	87.9 (71.4–95.5)	92.3 (88.5–94.9)
aOR (95% CI)				1.82 (0.60–5.52)	Reference	

Data are weighted and adjusted. Analyses among current smokers were adjusted for age, sex, income, education, country, and smoking frequency (daily vs. weekly). Analyses among ex-smokers were adjusted for age, sex, income, education, country, and length of quit duration (quit < 2 years ago vs. 2–5 years ago).

## Data Availability

In each country participating in the International Tobacco Control Policy Evaluation (ITC) Project, the data are jointly owned by the lead researcher(s) in that country and the ITC Project at the University of Waterloo. Data from the ITC Project are available to approved researchers 2 years after the date of issuance of cleaned data sets by the ITC Data Management Centre. Researchers interested in using ITC data are required to apply for approval by submitting an International Tobacco Control Data Repository (ITCDR) request application and subsequently to sign an ITCDR Data Usage Agreement. The criteria for data usage approval and the contents of the Data Usage Agreement are described online (http://www.itcproject.org, accessed on 26 September 2022).

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
