# Peer review of "Do Current Smokers and Ex-Smokers Who Use Nicotine Vaping Products Daily Versus Weekly Differ on Their Reasons for Vaping? Findings from the 2020 ITC Four Country Smoking and Vaping Survey"

_ijerph, 2022, doi:10.3390/ijerph192114130_

Round 1
Reviewer 1 Report
The article entitled "Do current smokers and ex-smokers who use nicotine vaping products daily versus weekly differ on their reasons for vaping? Findings from the 2020 ITC Four Country Smoking and Vaping Survey" is an interesting article.
I propose to take into account the following comments:
a) there is (lines 54-56):
"While NVPs are not harmless [1] and long-term risks are unknown [2], switching completely from smoking to vaping may reduce smoking-related diseases and death [3]."
but it is also known that vaping is harmful, for example, vaping for 30 minutes increases levels of oxidative stress, which can lead to lung damage, heart disease and Alzheimer's disease.
b) there is (lines 56-57):
"Additionally, there is 'moderate-certainty' evidence that NVPs can help people to stop smoking cigarettes [4]."
The use 'moderate-certainty' and 'can' in this sentence mean that NVPs help people to stop smoking cigarettes with less than 'moderate' certainty, i.e. "NVPs can help or cannot help".
In my opinion, there should be:
"Additionally, there is 'moderate-certainty' evidence that NVPs help people to stop smoking cigarettes." - but the reference to [4], after this correction, should be changed.
c) there is (lines 268-271, Conclusions):
"In all four countries, most respondents reported that NVPs may help them quit smoking or remain abstinent as a reason for vaping, particularly among those vaping daily."
But in accordance with the results presented in Table 1 ("Respondent Characteristics of the sample, Wave 1 (2020)") only a small number of adults who started vaping have quit smoking, i.e. large numbers of people continue to smoke, i.e. daily smoking: 57.3% (for Daily NVP use) and 62.8% (for Weekly NVP use).
Only a relatively small number of adults have quit smoking:
Former smoking (quit < 2 years): Daily NVP use (11.7%), Weekly NVP use (5.0%).
Former smoking (quit ≥ 2 years): Daily NVP use (16.7%), Weekly NVP use (1.9%).
The results show that adults begin to vape and continue smoking (saying they want to quit smoking).
Reviewer 2 Report
This MS is an update of previous ITC surveys. The authors provide an important data of reasons for vaping disaggregated by current cigarette smoker and past cigarette smokers. The list of reasons are very critical for health promotional and patient counselling purposes to educate the patients at the clinics or at bedside. There are some missing information in the MS in the methods and results/tables that has be filled up for better comprehension of general readers who do not have much information about ITC surveys. I presume the analyses only restricted to number of current and ex-smokers that was available. However, a brief introduction in methods total sample, sample design and prevalence and its 95% CI will be expected by readers. This would help reader judge if the country-level prevalence of smoking and e-cigarette prevalence had any influence on reasons for vaping. The analyses provides only unweighted but would it be necessary provide weighted estimates, and 95% CI of the reasons. There are % given in table 2 but 95%CI are for adjusted odds ratios.
The number of reasons for example is presented as mean ± perhaps is a standard deviation. SD should be within parenthesis but not as ± SD.
The analyses description provides steps taken to adjust for possible confounding by adding various demographics in each model. I would be interested to know the full results of each model. Please explain how outcome variable was classified for multinomial regression, were any factors (other than those shown in table 2) adjusted for in the models significant? Clarifications on the above is needed. Data analyses and presentation is not clear about regression analyses.
Reviewer 3 Report
The presented study disscuss reasons why adults who currently smoke or formerly smoked cigarettes use nicotine vaping products (NVPs) by vaping frequency stratified by smoking status. 3070 adults are included in this study from Australia, Canada, England, and United States. Authors found that both current and former smokers endorsed an average of six reasons for vaping, with those vaping daily reporting significantly more reasons than those vaping weekly. Among current smokers, 72.8% reported vaping may help them quit smoking, 13.0% reported vaping to reduce smoking but not to quit, and 14.2% reported vaping for reasons other than to reduce and/or quit smoking. The most common reason cited for vaping by respondents who formerly smoked was enjoyment, with those who vaped daily more likely than those who vaped weekly to report vaping for enjoyment and to reduce harm to themselves. In conclusion, Authors said a majority of respondents reported using NVPs to manage their smoking (reduce/quit smoking or remain quit), particularly those vaping daily. Those who vape daily also endorsed a greater variety of reasons other than managing smoking for using NVPs.
The topic is intersting and sutabile for publication in IJERPH
minor comments
According manuscript its A review or report but not article , i suggest to choose
Need to add some references and add more information in introduction
Need to present some figures instead of tables for more clarification
Round 2
Reviewer 2 Report
The revised manuscript read better as some analyses section queries were well answered and text was added in the R1 version to provide clarification to the readers. As mentioned that they have done many multiple regression models and only significant results were included in the manuscript table, authors may consider providing these models in an appendix for the benefit of readers who would want to read them.
Author Response
Reviewer:
The revised manuscript read better as some analyses section queries were well answered and text was added in the R1 version to provide clarification to the readers. As mentioned that they have done many multiple regression models and only significant results were included in the manuscript table, authors may consider providing these models in an appendix for the benefit of readers who would want to read them.
Authors' response: We have created two supplemental tables: (1) Supplemental Table 2: Logistic regression models on reasons for vaping among current smokers; and (2) Supplemental Table 3: Logistic regression models on reasons for vaping among ex-smokers.
We now refer to these tables on page 6 (Results for the (full) adjusted models on reasons for vaping among current smokers can be found in Supplemental Table 2) and on page 7 (Results for the (full) adjusted models on reasons for vaping among ex-smokers can be found in Supplemental Table 3.). The two supplemental files are attached.
